# AD-AutoGPT: An autonomous GPT for Alzheimer's disease infodemiology

**Haixing Dai**[1] , **Yiwei Li**[1] , **Zhengliang Liu**[1], **Lin Zhao**[1], **Zihao Wu**[1], **Suhang Song**[2], **Shen Ye**[3], **Dajiang Zhu**[4], **Xiang Li**[5,6], **Sheng Li**[7], **Xiaobai Yao**[8], **Lu Shi**[9], **Tai-Quan Peng**[10], **Quanzheng Li**[5,6], **Zhuo Chen**[2], **Donglan Zhang**[11], **Tianming Liu**[1], **Gengchen Mai**[8,12]*

**1** School of Computing, University of Georgia, Athens, Georgia, United States of America, **2** College of Public Health, University of Georgia, Athens, Georgia, United States of America, **3** Department of Epidemiology and Biostatistics, University of Georgia, Athens, Georgia, United States of America, **4** Department of Computer Science and Engineering, University of Texas at Arlington, Arlington, Texas, United States of America, **5** Massachusetts General Hospital, Boston, Massachusetts, United States of America, **6** Harvard Medical School, Boston, Massachusetts, United States of America, **7** School of Data Science, University of Virginia, Charlottesville, Virginia, United States of America, **8** Department of Geography, University of Georgia, Athens, Georgia, United States of America, **9** Department of Public Health Science, Clemson University, Clemson, South Carolina, United States of America, **10** Department of Communication, Michigan State University, East Lansing, Michigan, United States of America, **11** NYU Long Island School of Medicine, New York University, Mineola, New York, United States of America, **12** Department of Geography and the Environment, University of Texas at Austin, Austin, Texas, United States of America

☯ These authors contributed equally to this work.
* gengchen.mai@austin.utexas.edu

**Data availability statement:** The data, source code, and a demonstration video to showcase

## Abstract

In this pioneering study, inspired by AutoGPT, the state-of-the-art open-source application based on the GPT-4 large language model, we develop a novel tool called AD-AutoGPT, which can conduct data collection, processing, and analysis about complex health narratives of Alzheimer's Disease in an autonomous manner via users' textual prompts. We collated comprehensive data from a variety of news sources, including the Alzheimer's Association, BBC, Mayo Clinic, and the National Institute on Aging since June 2022, leading to the autonomous execution of robust trend analyses, intertopic distance map visualization, and identification of salient terms pertinent to Alzheimer's Disease. This approach has yielded not only a quantifiable metric of relevant discourse but also valuable insights into public focus on Alzheimer's Disease. This application of AD-AutoGPT in public health signifies the transformative potential of AI in facilitating a data-rich understanding of complex health narratives like Alzheimer's Disease in an autonomous manner, setting the groundwork for future AI-driven investigations in global health landscapes. Code, a demo video, and other information are available at https://github.com/levyisthebest/AD-AutoGPT.

## 1. Introduction

Alzheimer's Disease (AD), a progressive neurodegenerative disorder, remains one of the most pressing public health concerns globally in the 21st century [1,2]. It exerts an escalating burden on global healthcare systems as societies continue to age [3]. The significance of AD is

how to use our AD-AutoGPT for AD Infordemilogy are available through https://figshare.com/s/4e19cd8e63f58c11dca6.

**Funding:** DZ acknowledges support from the National Institutes of Health (NIH) through RO1 grant R01MD013886-05. DZ acknowledges partial support from grants R01AG075582 and RF1NS128534. SS acknowledges support from the UGA Interdisciplinary Research Pre-Seed Program—"Interdisciplinary Approaches to Alzheimer's Disease Prevention." XL acknowledges partial support from the National Institutes of Health through the grant "Identification of Multi-modal Imaging Biomarkers for Early Prediction of MCI-AD Conversion via Multigraph Representation" (1R03AG078625-01). We express our gratitude to the funding agencies for their financial support. The funders had no role in study design, data collection and analysis, decision to publish, or preparation of the manuscript.

**Competing interests:** The authors have declared that no competing interests exist.

further magnified by the increasing life expectancy globally, with the disease now recognized as a leading cause of disability and dependency among older people [4]. Consequently, AD has substantial social, economic, and health system implications, making its understanding and awareness of paramount importance [5,6].

Traditionally, public health professionals have to rely on labor-intensive methods such as web scraping, API data collection, data post-processing, and analysis/synthesis to gather insights from news media, health reports, and other textual sources [7–9]. However, these methods often necessitate complex pipelines for data gathering, processing, and analysis. Moreover, the sheer scale of global data presents an ever-increasing challenge, one that demands a novel, innovative approach to streamline these processes and extract valuable, actionable insights efficiently and automatically.

AutoGPT [10] is an experimental open-source application that harnesses the capabilities of large language models (LLMs) such as GPT-4 [11] and ChatGPT [12] to automate and optimize the analytical process. With its advanced linguistic understanding and autonomous operation, AutoGPT simplifies complex data pipelines [13,14], facilitating comprehensive analyses of vast datasets with simple textual prompts. In this study, inspired by the Auto-GPT architecture, we developed a robust autonomous LLM tool called AD-AutoGPT for Alzheimer's Disease Infodemiology. Fig 1 provides an illustration of the overall framework of our AD-AutoGPT. AD-AutoGPT were used to analyze a multitude of news sources, including the Alzheimer's Association, BBC, Mayo Clinic, and the National Institute on Aging, focusing on discourse since June 2022. We are among the pioneers in developing an autonomous LLM tool for public health informatics to elucidate the complex narrative surrounding Alzheimer's Disease. We summarize our key contributions below:

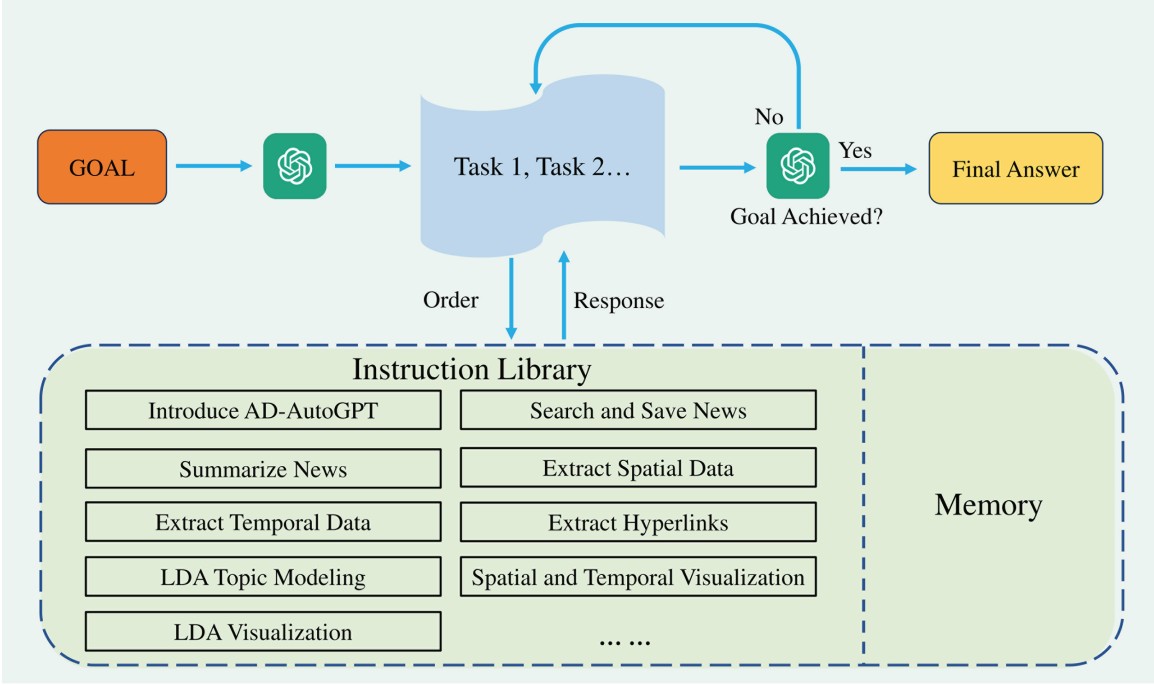

**Fig 1. The basic framework of AD-AutoGPT.**

- Inspired by AutoGPT, we develop a novel LLM-based tool called AD-AutoGPT, which can generate data collection, processing, and analysis pipeline in an autonomous manner based on users' textual prompts. More specifically, we adapt AD-AutoGPT to the public health domain to showcase its great potential of autonomous pipeline generation to understand the complex narrative surrounding Alzheimer's Disease.

- While AutoGPT is an effective autonomous LLM-based tool, it has lots of limitations when applying it on AD Infodemiology during the process of public health information retrieval, text-based information extraction, text summarization, summary analysis, and visualization. To overcome AutoGPT's limitations for the AD Infodemiology task, AD-AutoGPT provides the following improvements: (1) specific prompting mechanisms to improve the efficiency and accuracy of AD information retrieval; (2) a tailored spatiotemporal information extraction functionality; (3) an improved text summarization ability; (4) an in-depth analysis ability on generated text summaries; and (5) an effective and dynamic visualization capability.

- We show that AD-AutoGPT transforms the traditional labor-intensive data collection, processing, and analysis paradigm into a prompt-based automated, and optimized analytical framework.

- Through AD-AutoGPT, we have provided a case study for detailed trend analysis, intertopic distance mapping, and identified salient terms related to Alzheimer's Disease from four AD-related new sources. This contributes significantly to the existing body of knowledge and facilitates a nuanced understanding of the disease's discourse in public health.

- Our research underlines the capacity of AD-AutoGPT to facilitate data-driven public understanding of complex health narratives, such as Alzheimer's Disease, which is of paramount importance in an aging global society.

- The methodologies and insights from our work provide a foundation for future AI-assisted public health research. Our AD-AutoGPT pipeline is extendable to other topics in public health or even other domains. AD-AutoGPT is the first attempt to apply GPT automation technology to the field of AD. This innovative tool provides a powerful resource for AD research and diagnosis while also demonstrating the broader potential of this technology in medicine. Its success suggests that similar automated technologies could be extended to other diseases, such as Parkinson's, diabetes, and cardiovascular conditions, helping to accelerate research, improve diagnostic accuracy, and advance personalized medicine.

## 2. Background

### 2.1. Large language models

Large language models (LLMs) with their origins in Transformer-based pre-trained language models (PLMs) such as BERT [15] and GPT [16], have substantially transformed the field of natural language processing (NLP). LLMs such as GPT-3 [17], Bloom [18], PaLM [19], PaLM-2 [20], InstructGPT [21], Sparrow [22], ChatGPT [12], and GPT-4 [11], have superseded previous methods such as Recurrent Neural Network (RNN) based models, leading to their widespread adoption across various NLP tasks [12,23] including question answering, information extraction, sentiment analysis, text classification, and text generation.

Other than the applications in NLP domain [24], LLMs also show promising results and significant impacts in other disciplines such as biology [25], geography [26–35], earth system science [36], agriculture [37], education [38,39], medical and health care [40,41], and so on.

## 2.2. Public health infodemiology

Infodemiology [42] is a field that studies the determinants and distribution of information on the internet or in a population, with the goal of informing public health and public policy [9,42]. It plays a crucial role in monitoring and managing the information epidemic ("infodemic") associated with major public health crises.

Piamonte et al. [43] analyzed global search queries for AD using Google Trends data. They compared this online interest (Search Volume Index) with disease burden measures and revealed that search behavior and interest in AD were influenced by factors like news about celebrities with AD and awareness months.

With the rise of the internet and digital technologies, infodemiology provides a vital lens to examine the flow of health information and misinformation, helping public health practitioners develop effective communication strategies and interventions [44,45]. In the context of AD, understanding online behaviors and interests via infodemiology can help enhance public awareness, correct misconceptions, and inform preventative and management strategies for AD [43,46].

## 2.3. AutoGPT and LLM automation

The development and adoption of AutoGPT, LangChain[1], and many other automation techniques using LLMs mark a notable stride in NLP and artificial intelligence (AI). These frameworks like AutoGPT extend LLMs into prompt-instructed autonomous tools which provide a user-friendly interface even for non-expert users [10].

AutoGPT simplifies intricate tasks like data collection, cleaning, analysis, and even generating human-like text with straightforward prompts and eliminates the requirement for extensive coding or data science skills. This could democratize access to powerful LLMs, unlocking research and applications across diverse fields, including public health.

Recent studies [47,48] have highlighted the potential of autonomous LLM tools for automating the retrieval and analysis of large datasets. By crafting precise queries, AutoGPT can crawl various online platforms, gathering and analyzing discussions, comments, and posts about vaccines. It then generates a summarized report outlining major themes of public opinion and prevalent misconceptions, offering insights for public health officials to develop targeted communication and intervention strategies.

## 2.4. Improving autonomous LLM-based tools for public health

While recognizing the potential of autonomous LLMs like AutoGPT in public health research and practice, we identified certain limitations in their current state that may hinder their efficacy in particular use cases, such as infodemiology. By tailoring these tools to the specific needs of public health professionals, we aim to enhance their utility in these contexts.

Firstly, despite AutoGPT's extensive searching capabilities, it has limited abilities to acquire specialized information quickly and precisely. In response to this, we have integrated specific prompting mechanisms in our AD-AutoGPT. These tailored prompts direct AD-AutoGPT to gather data from a select list of authoritative websites relevant to AD.

Secondly, AD-AutoGPT also addresses the challenge AutoGPT faces in extracting critical details such as the time and place of events from news articles accurately. AD-AutoGPT uses web-crawling scripts to extract accurate timestamps from news pieces, and employs toponym recognition and resolution libraries such as geopy [49] and geopandas [50] to retrieve precise location information from texts.

---

[1] https://python.langchain.com/en/latest/index.html

Thirdly, depth of analysis is another area where AutoGPT needs further refinement. Owing to the token limit in models like ChatGPT, AutoGPT's analysis is often restricted to the first 4096 tokens [12]. Consequently, it might miss core content or important details. To overcome this limitation, AD-AutoGPT segments the text, vectorizes it, and then processes these chunks independently. It creates summaries for each of these segments and then amalgamates these summaries to create a comprehensive representation of the news article.

Fourthly, AutoGPT lacks the capacity to conduct an in-depth analysis of the generated summaries. AD-AutoGPT applies Latent Dirichlet Allocation (LDA) [51] to extract the most pertinent keywords from the text summaries, offering users a succinct understanding of the central themes in the AD domain.

Lastly, AutoGPT lacks robust visualization capabilities. Addressing this limitation, AD-AutoGPT integrates dynamic visualization techniques, creating plots of news occurrences over time, highlighting locations where news events are happening, and illustrating the evolution of research keywords over time.

Compared with AutoGPT, AD-AutoGPT is refined through the application of domain-specific knowledge and technical adjustments to optimize its relevance and effectiveness for public health researchers and practitioners.

## 3. Methods

In this section, we introduce AD-AutoGPT, an LLM-based tool we developed to automate the process of Alzheimer's Disease Infodemiology. AD-AutoGPT uses the Langchain framework to realize the connection with GPT-4 and ChatGPT API, and establish an LLM-based autonomous framework with a chain-of-thinking mode. AD-AutoGPT can automatically search for the latest news, extract meaningful spatio-temporal data, summarize the news, analyze news content, and visualize analysis results.

The overall framework of AD-AutoGPT is shown in Fig 1. We construct an instruction library that contains a set of possible commands/tools. A prompt shown in Fig 2 is designed to facilitate LLMs to identify usable tools from the instruction library and form a data processing pipeline that demonstrates the process of thinking. AD-AutoGPT's ability to "translate" natural language prompts to real data processing pipeline is similar to semantic parsing in traditional question answering [52–54], which aims at translating a natural language question into an executable query for a given knowledge base. The difference is that semantic parsing is only able to generate rather simple executable queries on a well-defined knowledge base while our AD-AutoGPT can handle much more complex real-world tasks such as searching and collecting news from Google, analyzing news contents, and visualizing analysis results.

### 3.1. Overall framework

Our primary goal is to learn from the chain thinking mode of AutoGPT to realize the automatic collection and summary of Alzheimer's disease news. The overall framework is shown in Fig 1. AD-AutoGPT will use ChatGPT or GPT-4 to divide the target task into several small tasks and process them separately. We provide AD-AutoGPT with an instruction library that contains customized functions/tools including:

1. *Introduce AD-AutoGPT*: Utilizing the power of LLMs, AD-AutoGPT can answer users' questions about AD-AutoGPT's basic information, such as introducing AD-AutoGPT's goals, functions and usage methods, etc.

2. *Search and Save News*: AD-AutoGPT utilizes Google API to search for the latest news posted on authoritative websites and save the URLs on a local device;

3. *Summarize News*: AD-AutoGPT employs ChatGPT or GPT-4 API to efficiently process large volumes of news text. After accessing saved news URLs and extracting text with web crawlers, it uses the LLMs for summarization.

4. *Extract spatial data*: AD-AutoGPT recognizes and extracts the geographic entities mentioned in each piece of news by using toponym recognition and resolution tools [55,56].

5. *Extract Temporal Data*: AD-AutoGPT extracts time-related data from news metadata;

6. *Extract Hyperlinks*: AD-AutoGPT identifies and extracts all the hyperlinks embedded within the news articles.

7. *LDA Topic Modeling*: AD-AutoGPT implements Latent Dirichlet Allocation (LDA) topic modeling to identify the topic distribution of news. This aids in distinguishing and categorizing news based on their content.

**A**

> Thought: I need to use the tool "Search and Save news" to find the latest news about Alzheimer's disease
> Action: Search and Save news
> Action Input: None

> Observation: News Links have been saved on this device.
> Thought: Now I can use the tool to summarize the news and draw plots based on the summary.
> Action: Summarize News
> Action Input: The file containing the news links.
> Browsing {website} and saving useful information in workplace folder…

> Observation: The news information you need is obtained, the summary information is stored under the workplace   folder and you can use them to get visualization results
> Thought: Now I can use the tool to draw plots based on the summary of the news.
> Action: Draw Plots
> Action Input: The summary information obtained from the previous step.
> Visualizing the news topics...
> Visualizing the news places...

> Observation: Every thing you need is obtained.
> Thought: Now I've got the final answer.
> Final answer: Based on the latest news about Alzheimer's disease, there have been some new studies on the potential benefits of exercise and a healthy diet in reducing the risk of developing Alzheimer's disease. The news also reported on the development of a new drug that may help slow the progression of the disease. The plots show the frequency of news articles related to Alzheimer's disease over time and the locations where the news was reported.

**Finished chain**
Based on the latest news about Alzheimer's disease, there have been some new studies on the potential benefits of exercise and a healthy diet in reducing the risk of developing Alzheimer's disease. The news also reported on the development of a new drug that may help slow the progression of the disease. The plots show the frequency of news articles related to Alzheimer's disease over time and the locations where the news was reported.

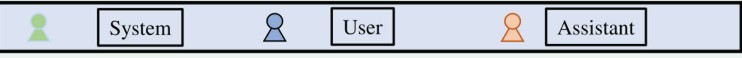

System   User   Assistant

**B**

Thought: I need to use the tool "Search and Save news" to find the latest news about Alzheimer's disease
Action: Search and Save news
Action Input: None

Observation: News Links have been saved on this device.
Thought: Now I can use the tool to summarize the news and draw plots based on the summary.
Action: Summarize News
Action Input: The file containing the news links.
Browsing {website} and saving useful information in workplace folder…

Observation: The news information you need is obtained, the summary information is stored under the workplace   folder and you can use them to get visualization results
Thought: Now I can use the tool to draw plots based on the summary of the news.
Action: Draw Plots
Action Input: The summary information obtained from the previous step.
Visualizing the news topics...
Visualizing the news places...

Observation: Every thing you need is obtained.
Thought: Now I've got the final answer.
Final answer: Based on the latest news about Alzheimer's disease, there have been some new studies on the potential benefits of exercise and a healthy diet in reducing the risk of developing Alzheimer's disease. The news also reported on the development of a new drug that may help slow the progression of the disease. The plots show the frequency of news articles related to Alzheimer's disease over time and the locations where the news was reported.

**Finished chain**
Based on the latest news about Alzheimer's disease, there have been some new studies on the potential benefits of exercise and a healthy diet in reducing the risk of developing Alzheimer's disease. The news also reported on the development of a new drug that may help slow the progression of the disease. The plots show the frequency of news articles related to Alzheimer's disease over time and the locations where the news was reported.

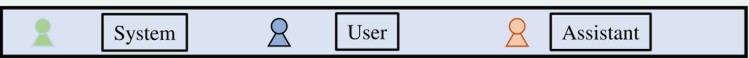

| System | | User | | Assistant |

**Fig 2. An example of AI thinking and calling functions to solve user problems.**

8. *Spatial and Temporal Visualization*: For spatial information, AD-AutoGPT will generate a map that highlights and showcases the mentioned locations from the news articles. For temporal information, AD-AutoGPT will construct a histogram depicting the monthly count of news reports on AD over a specific historical timeframe.

9. *LDA Visualization*: Based on the results of "LDA Topic Modeling", AD-AutoGPT represents topics by their highest-frequency and weightiest keywords for the user's interpretation. AD-AutoGPT will use the manifold diagram to show the overall topic trend over time based on the collected AD-related news.

After operating every small task choosing from these tools, AD-AutoGPT will judge whether the overall goal has been achieved according to the running results of the function, or it needs to think again and solve the next small problem. Chain thinking is realized through such a pattern.

### 3.2. Designing prompts to implement chain of thoughts

A prompt example can be seen in Fig 2a and the model thinking process of AD-AutoGPT is shown in Fig 2b. According to the input, this prompt has four parts in the task process which are question, thought, action, and action input.

1. *Question* is the problem that AI needs to solve.
2. *Thought* is the idea and thought process of AI for this problem.
3. *Action* is the operation which AI thinks is most suitable for solving the current task.
4. *Action input* is used as the input of the function.

For output, a prompt has three parts which are observation, thought, and final answer.

1. *Observation* is the output of the function to inspire AI's next thinking.
2. *Thought* shows the results of AI's thinking about *Observation*.
3. The *final answer* is the judgment of the result. If the AI thinks that the current result can fully answer the initial question, the AI will return the final answer. Otherwise, it will continue to think and call other functions.

The last part of the prompt is the question entered by the user, such as the question in Fig 2a, "*Can you help me to know something new about Alzheimer's Disease and maybe draw some plots for me?*". AI will decompose the complex target tasks proposed by users into several simple tasks, thus inspiring a chain of thoughts. And the thinking process of AI can be seen in Fig 2b.

Owing to this set of prompts, we can ensure that the thinking logic of AD-AutoGPT does not deviate from the right track and make the whole chain of thoughts visible to users.

### 3.3. Text summary

To achieve the purpose of extracting the most critical information from a large amount of news text, AD-AutoGPT performs new text summary.

The text summary is mainly achieved by accessing ChatGPT or GPT-4 API. Owing to the powerful text summarization ability of GPT-4, AD-AutoGPT can make more efficient use of text than other models. AD-AutoGPT traverses the saved news URLs one by one, and then saves the text from the website by calling the web crawler scripts. Next, it uses ChatGPT or GPT-4 to summarize the news text. All the text here will be pre-processed first, and then be summarized. More specifically, since GPT-4 has a limit on the number of tokens , we use the map_reduce method to process it [10].

In order to improve the trustworthiness of the results produced by AD-AutoGPT, AD-AutoGPT strictly restricts information retrieval to authoritative websites and designated trusted sources. These sources encompass verified medical databases, peer-reviewed scientific publications, and reputable medical websites. This approach ensures that the data processed and analyzed by the tool originates from reliable, high-quality sources, significantly minimizing the risk of producing inaccurate information. We also incorporate tools such as Deid-GPT [40] to perform anti-counterfeiting verification on the gathered information. These tools leverage advanced AI technology to detect and filter out potential fake data or inaccuracies, further enhancing the reliability of the generated content.

### 3.4. Spatiotemporal information extraction

Next, AD-AutoGPT will perform spatiotemporal information extraction on the collected news articles. Here, we adopt the geoparsing approach [55,57] which first recognizes place

names from raw text, so-called toponym recognition [26,28,56,58] and then link the recognized place names to a specific geographic entity in an existing gazetteer or geospatial knowledge graphs [59,60], so-called toponym resolution [61], so that the spatial footprints (i.e., geographic coordinates) of these places can be obtained. More specifically, we use GeoText[2], a python-based geoparsing tool to achieve this goal.

### 3.5. LDA analysis

Latent Dirichlet Allocation (LDA) [51] is a probabilistic topic model. LDA can give a probability distribution of topics of each document in the corpus. By analyzing a batch of document sets and extracting their topic distributions, topic clustering can be performed according to the topic distribution. LDA is a typical bag-of-words model, that is, a document is interpreted as a set of words, and there is no sequential relationship among words. In addition, a document can contain multiple topics, and each word in the document is assumed to be generated by one of the topics. LDA is an unsupervised learning method that does not require a manually labeled training set during training but only needs a document set and the total number of topics $K$. In addition, another advantage of LDA is that every topic is associated with a set of most frequent keywords that can be used to interpret this topic.

In short, AD-AutoGPT uses LDA topic modeling to discover the topics for the summary text of each piece of collected news, For each topic, the keyword with the highest frequency of occurrence and the highest weight will be displayed to the user.

## 4. Results

### 4.1. Alzheimer's disease news information retrieval

The effectiveness of our proposed AD-AutoGPT is mainly verified on the data provided by the most authoritative websites reporting Alzheimer's disease, which are Alzheimer's Association, BBC, National Institute of Aging, and Mayo Clinic. By using the prompt shown in Fig 2, we are able to instruct the LLM (e.g., ChatGPT or GPT-4) to search for the right tool in our instruction library—*Search and Save News* (see Fig 1) to achieve the first news data collection step.

We collected a total of 277 news articles from these four websites over the course of 2022 to 2023. On this news dataset, we validate the functions of AD-AutoGPT for text extraction, text summarization, spatio-temporal-data analysis, hot topics analysis, and result visualization. In this process, the time and location of the news will also be extracted and saved. Note that AD-AutoGPT automatically uses the given prompt and formalizes a data collection and processing pipeline based on the toolsets in our instruction library without any human intervention. Due to the high cost of OpenAI GPT-4, this study analyzed only 277 articles. However, with the assistance of AD-AutoGPT, researchers can update and analyze the latest articles in any quantity based on the provided data collection pipeline. Further details are available at [3].

### 4.2. Spatiotemporal information extraction and visualization

Based on the given prompt, AD-AutoGPT decides to use *Extract Spatial Data* tool and *Extract Temporal Data* tool in our instruction library (see Fig 1) to extract the places where these news articles mentioned and the timestamps when these news articles were posted online.

The spatial locations of extracted places from all news articles are visualized in Fig 3. Note that this map visualization is automatically generated by AD-AutoGPT based on the prompt

---

[2] https://github.com/elyase/geotext
[3] https://github.com/levyisthebest/AD-AutoGPT.

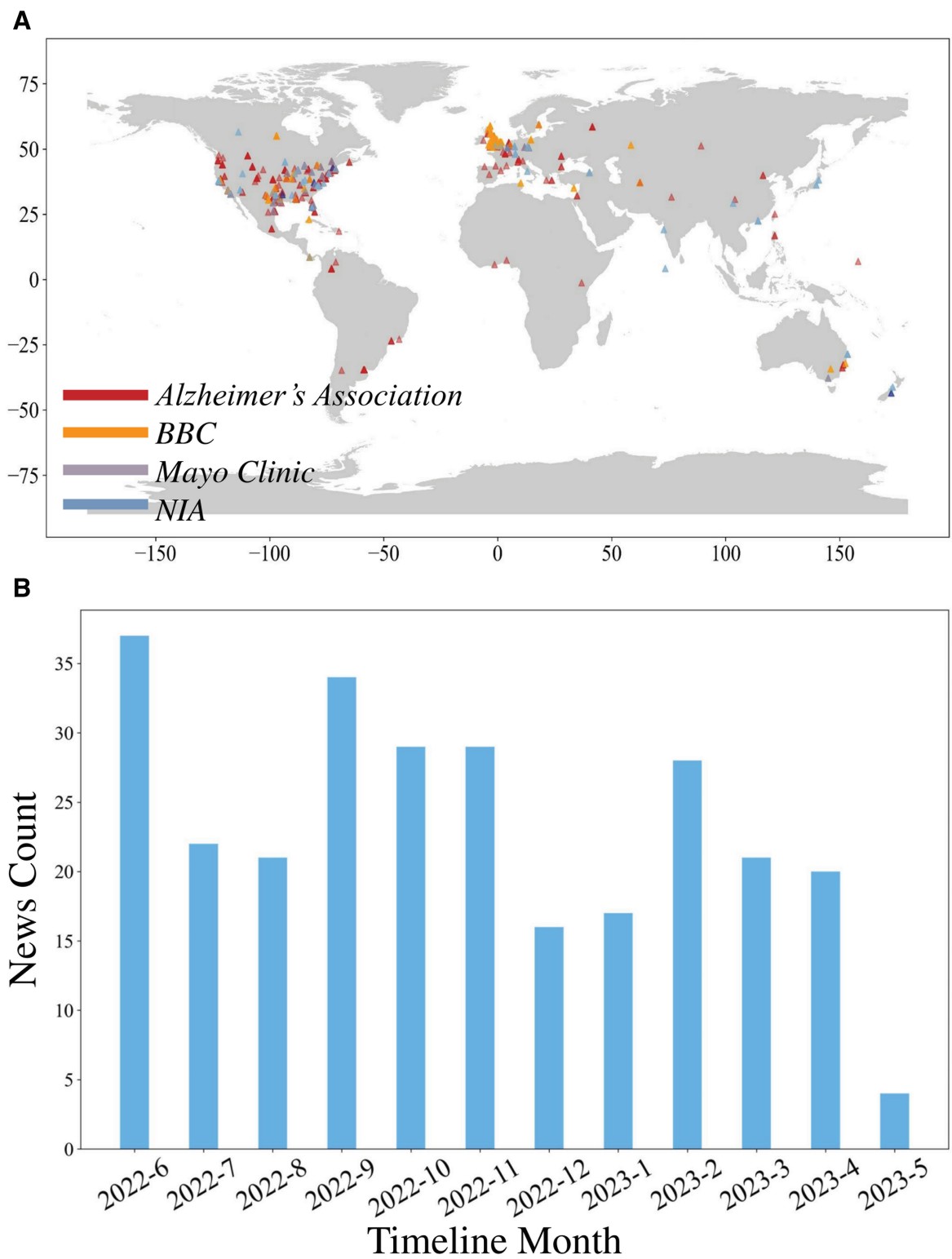

**Fig 3. a) Locations of recent Alzheimer's news; b) Monthly news count from June 2022 to May 2023.**

shown in Fig 2. For BBC news, although it primarily reports Alzheimer's disease news in the UK, the total number of news is not inferior to that of other websites. Similarly, websites in the United States such as NIA also pay more attention to local news, especially in the southeastern states of the United States. For the Alzheimer's Association, the sources of news reports are relatively scattered all over the world, while the United States and Western Europe still show higher report frequencies than other regions such as South America, Africa, Australia, and so on. Finally, for Mayo Clinic, since there is less news from this news source, only a few occurrences can be seen on the map. Generally speaking, the distribution of news is worldwide, but it is concentrated in the southeastern United States and Western Europe. This phenomenon might be a result of the selection bias from the four news media sources we used or the well-developed Alzheimer's disease research in these regions.

Temporal data analysis results can be seen in Fig 3b. It can be seen that the overall trend of the number of news reports is declining, from 31 in a single month in June 2022 to 13 in May 2023. It can also be seen that September, October, and November 2022 are the periods of high incidences of news reports. The number of news reports in each of the three months exceeded 27, and those in September 2022 reached 32, which was the highest in 2022. This might be because there was news that had a profound impact on AD-related media during this period, resulting in a sudden increase in reports, which deserves special attention from users.

We need to emphasize that these spatiotemporal analyses were done by AD-AutoGPT without any human input. Thereby AD-AutoGPT improves the efficiency of researchers' work, which AutoGPT cannot do because it does not design functions of information extraction from web pages.

## 4.3. LDA topic modeling and hot topic analysis

Based on the LDA topic modeling, a hot topic analysis is automatically conducted by AD-AutoGPT. The results can be seen in Fig 4. AD-AutoGPT aggregates the summaries of the news reported in the past year for LDA analysis, and finally got 5 hot topics. It selects the top 5 words with the most occurrences for each of the 5 hot topics and draws streamgraphs according to the number of occurrences and word weights of the words. Please refer to Fig 4a and 4b. In this way, you can understand the changes and trend of research topics.

It can be found that the keywords of the first hot topic (i.e., keywords indicated by pink boxes) are mainly protein, lipid, and drug, which shows that scientists are mostly concerned about seeking reliable drug treatment for Alzheimer's disease. The keywords of the topic with the second highest proportion (i.e., keywords indicated by yellow boxes) are individual, treatment, amyloid, and tissue. This topic is also about the drug treatment of Alzheimer's disease, but the focus has obviously shifted from the research and development of new drugs to the current personal medication, reflecting the patients' concerns about self-care. The keywords of the third-ranked topic i.e., keywords indicated by green boxes) include sleep, brain, blood, cell, etc. This type of news mainly focuses on the causes of Alzheimer's disease, which is similar to popular science news. It can be seen that journalists have attached great importance to popular science in the past year. For the fourth-ranked topic (i.e., keywords indicated by sky blue boxes), the keywords are increase, future, disorder, future, etc. This topic is mostly related to the future plan or expectation for Alzheimer's disease research. The keywords of the last topic are diagnosis, caregiver, vitamin, etc., reflecting the public's concerns about the diagnosis, care, and prevention of Alzheimer's disease.

Therefore, we can conclude that through hot topic analysis, we can easily get the popular topics in the news during the June 2022–May 2023 period by using AD-AutoGPT's

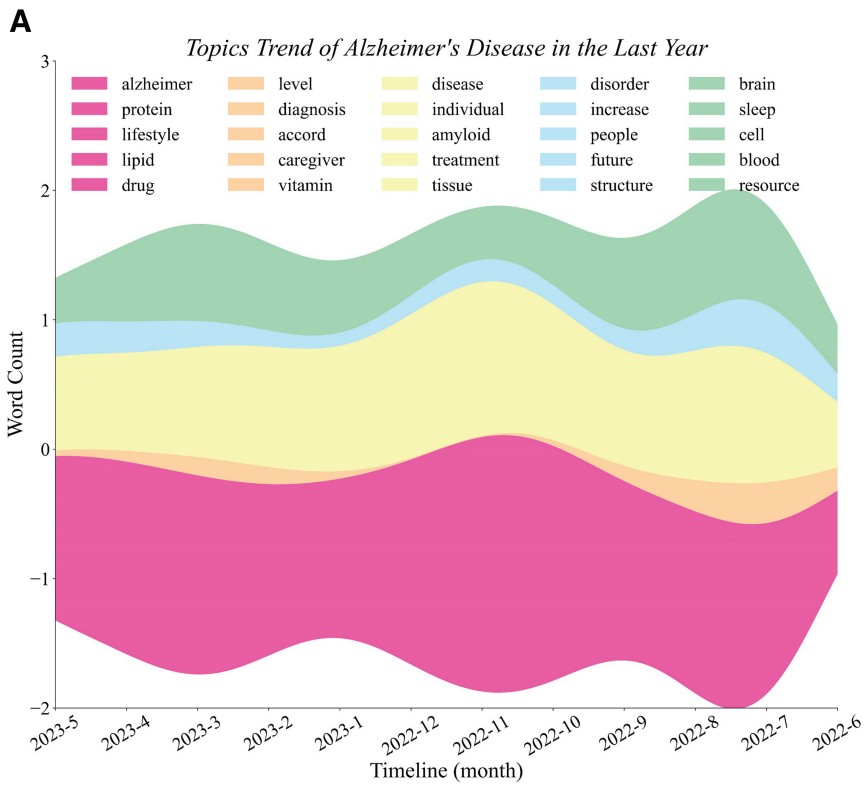

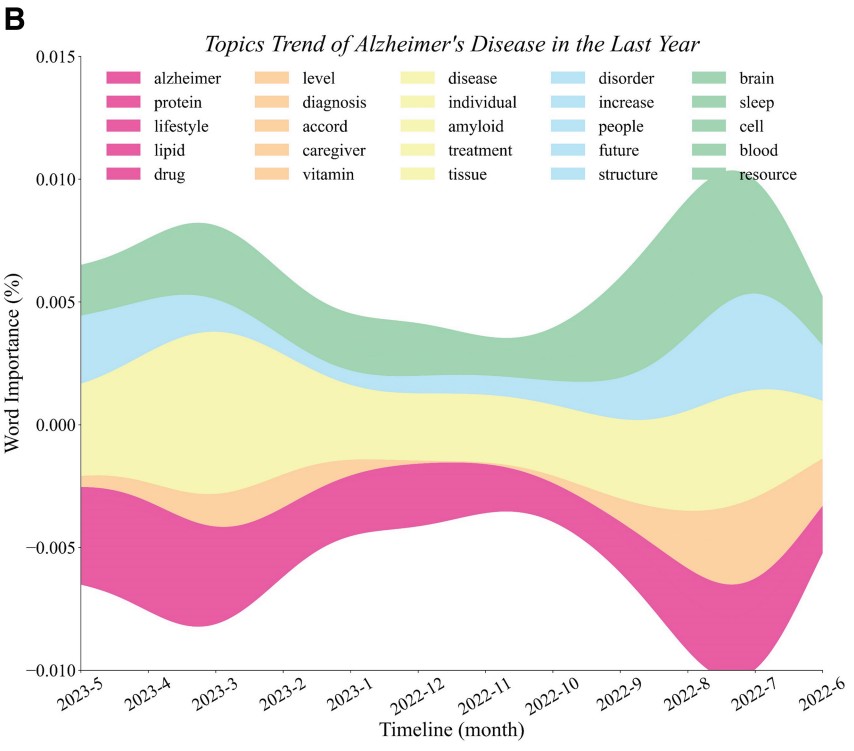

**Fig 4. Trends of word count and importance for each LDA-derived topic.**

autonomous workflow. Owing to GPT-4's powerful summarizing ability, in the future, the work of early information collection can be completely handed over to AI. Humans only need to judge and focus on the most critical information returned by AI to quickly understand the development and changes in the public health domain.

## 5. Discussion

### 5.1. Automating data analytics

The success of AD-AutoGPT shows the transformative potential of LLMs in the public health domain. By harnessing the advanced linguistic understanding and autonomous operations of AD-AutoGPT, we were able to streamline the analytical process and conduct comprehensive analyses of extensive news sources related to Alzheimer's Disease (AD). Moreover, AD-AutoGPT has the potential to go beyond the public health domain and be applied in various disciplines.

One of the key advantages of autonomous LLM-based tools such as AutoGPT and AD-AutoGPT is their ability to automate and optimize complex data extraction and analysis tasks, as well as transcending traditional labor-intensive methods. This enables researchers and professionals across different fields to access and engage with large language models, empowering them to conduct sophisticated analyses efficiently, regardless of their technical expertise.

### 5.2. Prioritizing insights and innovation

Through the development of AD-AutoGPT, we conducted a detailed trend analysis, intertopic distance mapping, and identified salient terms relevant to AD. By quantifying and visualizing the discourse, we gain a nuanced understanding of the prevalent topics, concerns, and perspectives related to AD, facilitating targeted interventions, communication strategies, and decision-making across multiple fields.

By automating data analysis tasks, researchers can dedicate more time and resources to interpreting the results and deriving actionable insights. This accelerates the research process and enhances the accuracy and reliability of the findings in diverse areas, such as social sciences, economics, technology, and more.

### 5.3. Transforming public health

Furthermore, the insights obtained from this research have broader implications beyond public health. The automation capabilities of AD-AutoGPT can revolutionize the field of infodemiology by efficiently analyzing online information trends, tracking the dissemination of information and misinformation, and predicting future trends.

While our AD-AutoGPT has made significant strides in utilizing autonomous LLM-based tools for AD analysis in the public health domain, there are still areas for further exploration and improvement. For example, based on different underlying pathologies, AD-related dementias (ADRD) can be categorized into four major types: prion disease, AD, frontotemporal lobar degeneration (FTLD), and Lewy body diseases (LBD). In practical clinical settings, differentiations among these subtypes of dementias are very challenging, due to both mixed pathologies and clinical symptoms. Our proposed AD-AutoGPT is a general framework and can be easily extended and refined to adapt to other dementias and various brain disorders. Additionally, exploring the integration of AD-AutoGPT with other data sources, such as social media platforms and electronic records, could provide a more holistic perspective on ADRD conversations and outcomes across multiple disciplines.

### 5.4. Ethical issues related to autonomous LLM-based tools

Utilizing autonomous LLM-based tools, several ethical issues arise that warrant careful consideration. First, these models generate output based on their training data, which if biased or discriminatory, could result in outputs that perpetuate such biases [62,63]. Ethical considerations must therefore include the selection and handling of training data of LLMs to minimize the risk of biased or inappropriate outputs.

In addition, issues of privacy and consent are paramount, particularly when dealing with sensitive data such as health information [40]. Even though LLMs do not remember specific inputs or retain personal data, the potential misuse of these tools can lead to leaking private or sensitive information, which raises significant ethical and legal questions.

Moreover, the potential for misuse extends to the propagation of false information or misinformation [64,65], a concern that is especially salient in the context of public health. LLMs can generate plausible-sounding but factually incorrect or misleading information [12,39].

Finally, the democratization of powerful technologies like AutoGPT also raises questions about responsibility and oversight. As these tools become more accessible and widespread, ensuring appropriate use and managing the potential for misuse becomes increasingly challenging.

## 6. Conclusion

In conclusion, this study proposes a transformative autonomous LLM-based tool called AD-AutoGPT which can facilitate data-driven understanding of complex narratives, not limited to public health but also applicable to various other domains. The initial success of AD-AutoGPT has laid the foundation for the future LLM-assisted investigations in global health landscapes and beyond. We believe that our approach has significant potential to be reused in other disease areas to make a meaningful impact. By leveraging the power of large language models and automation techniques, researchers and professionals can gain valuable insights, inform evidence-based interventions, and drive positive impact across diverse domains.

## Author contributions

**Conceptualization:** Haixing Dai, Zhengliang Liu, Lin Zhao, Zihao Wu, Suhang Song, Shen Ye, Dajiang Zhu, Xiang Li, Sheng Li, Xiaobai Yao, Lu Shi, Tai-Quan Peng, Quanzheng Li, Zhuo Chen, Donglan Zhang, Gengchen Mai, Tianming Liu.

**Data curation:** Haixing Dai, Yiwei Li, Zhengliang Liu, Lin Zhao, Zihao Wu, Tai-Quan Peng.

**Formal analysis:** Haixing Dai, Yiwei Li.

**Funding acquisition:** Suhang Song, Zhuo Chen, Donglan Zhang, Gengchen Mai, Tianming Liu.

**Investigation:** Tianming Liu.

**Methodology:** Haixing Dai, Yiwei Li, Zhengliang Liu, Lin Zhao, Zihao Wu, Suhang Song, Dajiang Zhu, Sheng Li, Tai-Quan Peng, Zhuo Chen, Donglan Zhang, Gengchen Mai, Tianming Liu.

**Project administration:** Gengchen Mai, Tianming Liu.

**Resources:** Xiang Li.

**Software:** Haixing Dai, Yiwei Li, Zhengliang Liu, Tianming Liu.

**Supervision:** Shen Ye, Gengchen Mai, Tianming Liu.

**Validation:** Xiang Li, Xiaobai Yao, Lu Shi, Quanzheng Li, Gengchen Mai, Tianming Liu.

**Visualization:** Haixing Dai, Gengchen Mai.

**Writing – original draft:** Haixing Dai, Yiwei Li, Zhengliang Liu, Dajiang Zhu, Tianming Liu.

**Writing – review & editing:** Haixing Dai, Yiwei Li, Zhengliang Liu, Lin Zhao, Zihao Wu, Suhang Song, Shen Ye, Xiang Li, Sheng Li, Xiaobai Yao, Lu Shi, Tai-Quan Peng, Quanzheng Li, Zhuo Chen, Donglan Zhang, Gengchen Mai, Tianming Liu.

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
