## [Decision Letter · Decision Letter 0]

5 Aug 2024

PGPH-D-24-00508

AD-AutoGPT: An Autonomous GPT for Alzheimer’s Disease Infodemiology

Dear Dr. Mai,

Thank you for submitting your manuscript to PLOS Global Public Health. After careful consideration, we feel that it has merit but does not fully meet PLOS Global Public Health’s publication criteria as it currently stands. Therefore, we invite you to submit a revised version of the manuscript that addresses the points raised during the review process.

Please see the comments of two reviewers below. Both reviewers have raised some initial, large-scale queries that we now invite you to address.

We look forward to receiving your revised manuscript.

Kind regards,

Hanna Landenmark

Staff Editor

Journal Requirements:

1. We have amended your Competing Interest statement to comply with journal style. We kindly ask that you double check the statement and let us know if anything is incorrect.

2. Figure 3: please (a) provide a direct link to the base layer of the map (i.e., the country or region border shape) and ensure this is also included in the figure legend; and (b) provide a link to the terms of use / license information for the base layer image or shapefile. We cannot publish proprietary or copyrighted maps (e.g. Google Maps, Mapquest) and the terms of use for your map base layer must be compatible with our CC-BY 4.0 license. 

Additional Editor Comments (if provided):

Reviewers' comments:

Reviewer's Responses to Questions

**Comments to the Author**

1. Does this manuscript meet PLOS Global Public Health’s publication criteria? Is the manuscript technically sound, and do the data support the conclusions? The manuscript must describe methodologically and ethically rigorous research with conclusions that are appropriately drawn based on the data presented.

Reviewer #1: Yes

Reviewer #2: Yes

2. Has the statistical analysis been performed appropriately and rigorously?

Reviewer #1: Yes

Reviewer #2: No

3. Have the authors made all data underlying the findings in their manuscript fully available (please refer to the Data Availability Statement at the start of the manuscript PDF file)?

Reviewer #1: Yes

Reviewer #2: No

4. Is the manuscript presented in an intelligible fashion and written in standard English?

Reviewer #1: Yes

Reviewer #2: Yes

5. Review Comments to the Author

Reviewer #1: This paper introduces AD-AutoGPT, an advanced Large Language Model (LLM) tool designed to collect, process, and analyze data based on users' textual prompts, specifically focusing on understanding complex health narratives related to Alzheimer’s disease. While AD-AutoGPT holds significant promise for enhancing our comprehension of AD-related health issues, it's crucial to acknowledge potential risks related to the dissemination of misinformation and its impact on public health. Are there any risk mitigation processes or safeguards employed to prevent the mass generation of health disinformation?

Reviewer #2: Major:

1. While I find the tool presented useful, the authors should tone down a bit their claims through the text. There are sentences like: "The initial success of AD-AutoGPT has paved the way for future..."

2. The state of the art is not discussed in the introduction. Is there the first disease-specific application of AutoGPT? Are there other AD-AutoGTP already? If so, cite them, explain what's different between your tool and these. If not, the authors should emphasize the reproducibility aspect much more since their work can be reused for other diseases.

3. Related to the last point, I would like to see the code in GitHub with clear instructions and a link to figshare for data/videos. The authors should emphasize how someone can reuse their work for any other indication with the minimum effort.

4. The samples size seems small (+200 articles). Why not adding many more? I feel a tool with only 200 articles between 2022 and 2023 is outdated for any research. How do the authors plan to keep the tool up to date? Arent there ways to automatize including more articles to it?

6. PLOS authors have the option to publish the peer review history of their article (what does this mean?). If published, this will include your full peer review and any attached files.

**Do you want your identity to be public for this peer review?** For information about this choice, including consent withdrawal, please see our Privacy Policy.

Reviewer #1: No

Reviewer #2: No

---

## [Decision Letter · Decision Letter 1]

22 Oct 2024

PGPH-D-24-00508R1

AD-AutoGPT: An Autonomous GPT for Alzheimer’s Disease Infodemiology

Dear Dr. Mai,

Thank you for submitting your manuscript to PLOS Global Public Health. After careful consideration, we feel that it has merit but does not fully meet PLOS Global Public Health’s publication criteria as it currently stands. Therefore, we invite you to submit a revised version of the manuscript that addresses the points raised during the review process.

The manuscript has been evaluated by two reviewers, and their comments are available below.

Although reviewer 1 is satisfied with the revised manuscript, reviewer 2 still has some concerns. The reviewer requests:

1) A more complete github repository

2) Further details regarding the sample of articles. The reviewer's original concern was:

"The samples size seems small (+200 articles). Why not adding many more? I feel a tool with only 200 articles between 2022 and 2023 is outdated for any research. How do the authors plan to keep the tool up to date? Aren't there ways to automatize including more articles to it?"

Your reply was:

"We acknowledge the reviewer’s concerns regarding the sample size. Our current dataset includes just

over 200 articles from 2022 and 2023. This limitation is primarily due to our stringent keyword search

rules, which reduced the number of qualifying results. Additionally, at the time of our study, the cost of

using GPT-4 was relatively high, and our funding constraints prevented us from gathering a larger dataset.

Nevertheless, as a methodology-focused paper, we hope that this sample size is sufficient to demonstrate the

validity of our approach. Future researchers can use our tool to explore a broader dataset, which we are

confident will yield even more meaningful insights."

Although this response addresses the first half of the reviewer's question, I do not think it deals with the second half: "How do the authors plan to keep the tool up to date? Aren't there ways to automatize including more articles to it?"

Could you please carefully revise the manuscript to address all comments raised?

We look forward to receiving your revised manuscript.

Kind regards,

Steve Zimmerman, PhD

PLOS Staff Editor

Journal Requirements:

Additional Editor Comments (if provided):

Reviewers' comments:

Reviewer's Responses to Questions

**Comments to the Author**

1. If the authors have adequately addressed your comments raised in a previous round of review and you feel that this manuscript is now acceptable for publication, you may indicate that here to bypass the “Comments to the Author” section, enter your conflict of interest statement in the “Confidential to Editor” section, and submit your "Accept" recommendation.

Reviewer #1: All comments have been addressed

Reviewer #2: (No Response)

2. Does this manuscript meet PLOS Global Public Health’s publication criteria? Is the manuscript technically sound, and do the data support the conclusions? The manuscript must describe methodologically and ethically rigorous research with conclusions that are appropriately drawn based on the data presented.

Reviewer #1: Yes

Reviewer #2: Partly

3. Has the statistical analysis been performed appropriately and rigorously?

Reviewer #1: N/A

Reviewer #2: N/A

4. Have the authors made all data underlying the findings in their manuscript fully available (please refer to the Data Availability Statement at the start of the manuscript PDF file)?

Reviewer #1: Yes

Reviewer #2: No

5. Is the manuscript presented in an intelligible fashion and written in standard English?

Reviewer #1: Yes

Reviewer #2: Yes

6. Review Comments to the Author

Reviewer #1: (No Response)

Reviewer #2: The authors still did not fully allow for reproducing the study.

For example, the github repository readme indicates that to install the repo, you have to run pip install -r reqiurements.txt (which has a typo btw). However, there is not such file there (https://github.com/levyisthebest/AD-AutoGPT).

The files are simply dump there, without any explanation, no notebooks to reproduce the figures, etc.

I cannot see the reply from the authors to this comment from the previous version.

> 4. The samples size seems small (+200 articles). Why not adding many more? I feel a tool with only 200 articles between 2022 and 2023 is outdated for any research. How do the authors plan to keep the tool up to date? Arent there ways to automatize including more articles to it?

7. PLOS authors have the option to publish the peer review history of their article (what does this mean?). If published, this will include your full peer review and any attached files.

**Do you want your identity to be public for this peer review?** For information about this choice, including consent withdrawal, please see our Privacy Policy.

Reviewer #1: No

Reviewer #2: No

---

## [Editor Report · Decision Letter 2]

18 Feb 2025

AD-AutoGPT: An Autonomous GPT for Alzheimer’s Disease Infodemiology

PGPH-D-24-00508R2

Dear Dr. Mai,

We are pleased to inform you that your manuscript 'AD-AutoGPT: An Autonomous GPT for Alzheimer’s Disease Infodemiology' has been provisionally accepted for publication in PLOS Global Public Health.

Best regards,

Somayeh Hessam

Academic Editor